# Regulation of ADAM10 by the TspanC8 Family of Tetraspanins and Their Therapeutic Potential

**DOI:** 10.3390/ijms22136707

**Published:** 2021-06-23

**Authors:** Neale Harrison, Chek Ziu Koo, Michael G. Tomlinson

**Affiliations:** 1School of Biosciences, University of Birmingham, Birmingham B15 2TT, UK; n.harrison@bham.ac.uk (N.H.); CXK543@bham.ac.uk (C.Z.K.); 2Centre of Membrane Proteins and Receptors (COMPARE), Universities of Birmingham and Nottingham, Midlands, UK

**Keywords:** metalloproteinase, ADAM10, tetraspanin, TspanC8, membrane protein, ectodomain shedding

## Abstract

The ubiquitously expressed transmembrane protein a disintegrin and metalloproteinase 10 (ADAM10) functions as a “molecular scissor”, by cleaving the extracellular regions from its membrane protein substrates in a process termed ectodomain shedding. ADAM10 is known to have over 100 substrates including Notch, amyloid precursor protein, cadherins, and growth factors, and is important in health and implicated in diseases such as cancer and Alzheimer’s. The tetraspanins are a superfamily of membrane proteins that interact with specific partner proteins to regulate their intracellular trafficking, lateral mobility, and clustering at the cell surface. We and others have shown that ADAM10 interacts with a subgroup of six tetraspanins, termed the TspanC8 subgroup, which are closely related by protein sequence and comprise Tspan5, Tspan10, Tspan14, Tspan15, Tspan17, and Tspan33. Recent evidence suggests that different TspanC8/ADAM10 complexes have distinct substrates and that ADAM10 should not be regarded as a single scissor, but as six different TspanC8/ADAM10 scissor complexes. This review discusses the published evidence for this “six scissor” hypothesis and the therapeutic potential this offers.

## 1. Introduction

The evolutionary conserved superfamily of Zn^2+^ dependent a disintegrin and metalloproteinases (ADAMs) are single pass transmembrane metalloproteinases that are responsible for the ectodomain shedding of a large number of transmembrane proteins. This “molecular scissor” action can have dramatic effects on cell function, including the initiation of intracellular signaling pathways, downregulation of receptor signaling or adhesive function, or the release of soluble mediators such as growth factors, cytokines, or chemokines [1]. ADAMs share a common structure, comprising an N-terminal signal sequence followed by an inhibitory prodomain, a metalloproteinase domain, disintegrin domain, cysteine-rich region, an epidermal growth factor (EGF) domain (missing in ADAM10 and 17), a transmembrane region, and a cytoplasmic tail. The inhibitory prodomain is removed by proprotein convertases, such as furin, in the Golgi to generate a mature, active metalloproteinase [2]. Of the 22 ADAM genes currently identified in humans, 12 are catalytically active but 10 are catalytically inactive and remain poorly characterized [3]. ADAM10 and ADAM17 are the best studied, largely due to knockout mouse studies that have highlighted their important roles in embryonic development and response to injury/infection, respectively. ADAM10 is essential for the cleavage and activation of Notch cell fate determining proteins to enable embryonic development [4,5,6]. ADAM17, also known as tumor necrosis factor α (TNFα) converting enzyme (TACE), is important in first line of defense by releasing TNFα and EGF receptor ligands [7]. ADAM10 and ADAM17 appear to function as molecular scissor complexes with regulatory tetraspanins and inactive rhomboids (iRhoms), respectively [8,9,10]. This review focuses on the regulation of ADAM10 by tetraspanins.

## 2. Importance of the Molecular Scissor ADAM10 in Health and Disease

ADAM10 knockout mice die at embryonic day 9.5 with defects of the cardiovascular system, neuronal system, and somites, thus phenocopying Notch1/4 double knockout mice [5]. Subsequent conditional knockout of ADAM10 in specific cell types have shown its importance in many different developmental processes including brain development and neurogenesis, vascular development, and epidermal morphogenesis and homeostasis [11,12,13]. The role of ADAM10 in many of these developmental processes appears to be due to its activation of Notch proteins, of which there are four family members, Notch1–4. ADAM10 is essential for cleaving Notch proteins at the S2 proteolytic cleavage site upon binding of Delta-like or Jagged ligands expressed on the surface of other cells. This allows further cleavage by γ-secretase at the S3 site to release the Notch intracellular domain (NICD), which then translocates to the nucleus to induce transcription of its target genes [14]. ADAM10 has at least 100 other substrates [15], including cadherins [16,17,18], and the EGF receptor ligands EGF and betacellulin [19], which are important in cell adhesion and embryonic morphogenesis. The number of ADAM10 substrates is likely to rise substantially as future studies determine the ADAM10 “sheddome” in wild-type versus ADAM10 knockouts for different cell types using mass spectrometry proteomics.

The role of certain ADAM10 substrates in disease progression suggests that ADAM10 is a promising therapeutic target, with notable possibilities for Alzheimer’s disease, cancer, and inflammatory diseases [20,21]. In Alzheimer’s disease, ADAM10 activation has the potential to inhibit disease progression, because ADAM10 cleavage of amyloid precursor protein (APP) prevents the generation of the pathogenic amyloid β peptide, which can otherwise be produced by β- and γ-secretases cleaving at alternative sites [22]. Indeed, over-expression of ADAM10 in the neurons of a mouse Alzheimer’s disease model reduces amyloid β production and deposition in plaques, while over-expression of a catalytically inactive ADAM10 has the opposite effect [22]. Another ADAM10 substrate for which shedding is predicted to protect against disease is the platelet-activating collagen/fibrin/fibrinogen receptor GPVI [23,24]. GPVI deficiency protects against arterial thrombosis in mouse models, without causing any major bleeding side-effects, and a GPVI blocking antibody (ACT017) induces disaggregation of platelet thrombi [25]. The ACT017 antibody is the subject of a phase I clinical trial for patients suffering from acute ischaemic stroke (NCT03803007, www.clinicaltrials.gov (accessed on 21 May 2021)).

In many other diseases, ADAM10 activity clearly promotes the disease. For example, T-cell acute lymphoblastic leukemia (T-ALL) is an aggressive blood cancer that is in urgent need of more effective treatments, and approximately two thirds of T-ALL is driven by activating mutations in Notch1 that allow its activation by ADAM10 without the need for normal ligand binding. The potential for ADAM10 inhibition as a therapeutic strategy has been demonstrated by ADAM10 knockdown or inhibition in T-ALL cell lines, which inhibits constitutive Notch signaling and reduces cell proliferation [26]. A second example is breast cancer driven by over-expression of the EGF receptor family member HER2. This cancer is treated by targeting HER2 with the antibody Herceptin, but one resistance mechanism to the therapy appears to be upregulation of ADAM10 and its cleavage and removal of HER2 [27,28]. Combination treatment of breast cancer cell lines with Herceptin and ADAM10 knockdown or inhibition potentiates the cancer-inhibiting effect of the antibody [29], suggesting that such combination therapy might overcome Herceptin resistance in patients. Finally, inhibition of ADAM10 shedding of CD23, the low affinity receptor for immunoglobulin E (IgE) on B cells, has been proposed as a potential therapy for asthma, since specific deletion of ADAM10 in B cells is protective against asthma in a mouse model [30].

There has been considerable historical interest in ADAM10 as a therapeutic target, but there are no approved ADAM10 therapies yet. Most of the ADAM10 therapeutics that have reached clinical trials to date are not targeted to ADAM10 directly, but rather target other proteins that in turn promote ADAM10 expression or maturation. Such drugs are focused on Alzheimer’s disease and have shown neuroprotective efficacy in cell line and mouse models, but clinical trials have not progressed beyond phase II [21]. Direct targeting of ADAM10 has concentrated on the use of hydroxamate class small molecule inhibitors, in particular GI254023X, which has 100-fold higher potency in inhibiting ADAM10 than ADAM17 [31]. GI254023X is widely used in academic research, but has no clinical trials listed on the ClinicalTrials.gov registry. There is one ADAM10-related trial in phase I for the treatment of high-grade gliomas in children (NCT04295759, www.clinicaltrials.gov (accessed on 21 May 2021)), but the inhibitor in question, INCB7839, inhibits both ADAM10 and ADAM17 [32]. Given that ADAM10 is expressed by most, if not all, cells in the body, it is likely that inhibition of this important enzyme will result in toxic side effects. Therefore, there is a current need to better understand ADAM10 regulation which may enable its targeting in a cell type- or substrate-specific manner, so minimizing toxicity.

## 3. Tetraspanins as Regulators of Other Membrane Proteins

### 3.1. Tetraspanin Overview

Tetraspanins are a superfamily of proteins with four transmembrane domains, two extracellular regions and intracellular N- and C-termini (Figure 1) that appear to be present in most multicellular organisms. They function by interacting with specific partner proteins and regulating their intracellular trafficking, lateral mobility, and clustering at the cell surface [33]. There are 33 human tetraspanins, systematically named Tspan1–Tspan33, although cluster of differentiation (CD) nomenclature is more commonly used for those with CD names. A full list of alternative genes names for the tetraspanins can be found at the HUGO Gene Nomenclature Committee website (https://www.genenames.org/ (accessed on 21 May 2021)). Tetraspanins interact with each other and a range of other membrane partner proteins including adhesion and signaling receptors, and ectoenzymes such as ADAM10. Each cell type expresses its own complement of tetraspanins, with most expressing at least 20 [34]. Super resolution microscopy suggests that tetraspanins exist in the plasma membrane as small assemblies, or nanodomains, composed of approximately 10 tetraspanin molecules of a single type per cluster [35,36,37].

### 3.2. Tetraspanins Interact with and Regulate Other Membrane Proteins

The main role of tetraspanins is to bind to their specific partner proteins and regulate their intracellular trafficking, lateral mobility in the membrane, and clustering into nanodomains. Notable examples of this regulatory role include the interaction of tetraspanin CD151 with the laminin-binding integrins α3β1 and α6β1 [42], tetraspanin CD81 with the B-cell costimulatory receptor CD19 [39], Tspan12 with the Wnt/Norrin receptor Frizzled 4 [43], Tspan18 with the store-operated Ca^2+^ entry channel Orai1 [44,45], and tetraspanin CD9 with EWI-2 and EWI-F, two related immunoglobulin superfamily proteins of unknown function [40,46]. Importantly, the dependence of partner proteins on the tetraspanin in these examples is evidenced by the fact that the phenotypes of tetraspanin-deficient cell lines or animals mimic partner protein deficiency, and/or that recent structural studies have demonstrated the intimate association of tetraspanins with partners. There are many other tetraspanin/partner protein interactions reported in the literature, often detected by co-immunoprecipitation in detergent lysates. It will be important to assess the relevance of such interactions using the functional or structural validation approaches described for the examples above.

### 3.3. Insights from Recent Tetraspanin Structural Studies

Tetraspanins differ from other proteins containing four transmembrane domains in that they share conserved amino acids, such as conserved cysteine residues in the main extracellular domain, including a cysteine-cysteine-glycine motif, which form disulphide bridges important for the domain’s structure. The first full-length crystal structure of a tetraspanin was solved for CD81 in 2016 by the Blacklow group [38]. CD81 was shown to have a cone-shaped structure with an intramembrane cavity (Figure 1). A cholesterol molecule occupied the cavity, and molecular dynamics simulations suggested that loss of cholesterol would allow the main extracellular domain to swing upwards, resulting in a dramatic conformational change from “closed” to “open” conformation [38]. The Blacklow group extended this line of work by recently reporting the cryo-electron microscopy (cryo-EM) structure of CD81 in complex with CD19 and the antibody-binding fragment (Fab) of a CD19 antibody [39]. This showed CD81 in the extended, open conformation and the CD19 ectodomain resting on top of CD81, with these ectodomains providing the main binding interface [39]. Rearrangements of the CD81 transmembranes in the CD19-bound conformation narrow the intramembrane cavity, preventing cholesterol from binding (Figure 1). The authors suggest a model whereby the extended CD81/CD19 complex forms in the cholesterol-low environment of the endoplasmic reticulum, before trafficking to the B cell surface to function as a costimulatory complex for B cell receptor-induced activation [39]. The complex appears to dissociate upon B cell receptor signaling, presumably to allow engagement of CD19 with the B cell receptor and CD81′s return to the closed, cholesterol-bound conformation in the relatively cholesterol-rich environment of the plasma membrane [47].

Two additional crystal structures of tetraspanins were recently reported, namely for CD9 [40] and CD53 [41]. The cone-shaped structures created by the transmembranes are similar for CD9, CD53, and CD81, suggesting a basic structure that may be common for all tetraspanins (Figure 1). An additional feature of the CD9 and CD53 structures is the visualization of the small extracellular region, on which the larger extracellular region rests to close off the transmembrane cavity. Within this cavity, cholesterol was not detected; instead, the lipid species used to reconstitute the purified tetraspanins for crystallization was present [40,41]. This supports the hypothesis for lipid binding within the cavity, but it remains to be determined whether all tetraspanins in the native state bind cholesterol or alternative lipids. The closed position of the CD9 major extracellular region is similar to CD81 [40], which is not surprising given that these tetraspanins are relatively closely related by amino acid sequence (45% identity). This region adopts the mushroom-like structure that was previously reported for the CD81 extracellular region in isolation [48], with two alpha helices forming the stalk (A and E helices) and three forming the head (B, C, and D). A difference in CD9 is a loop rather than a helix in the D region, and a more loop-like C region [40,46]. The CD53 main extracellular region exhibits additional differences, existing in a more extended, open conformation and having loops rather than helices for the C and D regions [41]. Such differences in the C and D regions are unsurprising given that CD53 is not particularly closely related to CD9 and CD81 (27% and 31% amino acid sequence identity, respectively).

Two further cryo-EM structures of tetraspanins in complex with partner proteins were recently reported, namely CD9 with its partners EWI-F or EWI-2 [40,46]. These were relatively low resolution structures and included CD9 Fab or nanobodies, but both showed a hetero-tetrameric complex with centrally dimerized EWI proteins flanked by CD9 proteins on either side, with some flexibility in the position of each CD9 [40,46]. The interactions appeared to be mediated by the transmembrane regions of the proteins [40,46], consistent with previous co-immunoprecipitation data using mutant constructs [49]. The main extracellular region of the two CD9 proteins adopted an extended, open conformation in both structures [40,46].

These structures have provoked some interesting hypotheses related to tetraspanin function. First, the observed flexibility in the structure of the D region of the major extracellular region, for example in complex with different nanobodies, suggests that it can adopt different conformations to bind different partners [46], helping to explain how a single tetraspanin binds different partners. Second, a comparison of the extended conformation of CD81 with a model of extended CD9 demonstrates how two relatively similar tetraspanins have different binding specificities. This is because a key CD19-binding loop on the C region is hydrophobic for CD81, but is hydrophilic for the non-binding CD9 [39]. Third, the cone-shaped structure of tetraspanins may induce membrane curvature and help to explain the relatively high expression of tetraspanins on extracellular vesicles [40]. Fourth, since tetraspanins have previously been proposed to homodimerize, the hetero-tetramers of CD9 and EWI proteins may link to form chains to help to explain how tetraspanins form clusters within nanodomains [46]. Finally, the capacity of the main extracellular region to adopt closed and open conformations suggests that tetraspanins may regulate some partners as “molecular switches” via conformational change in a cholesterol-regulated manner.

Non-conventional tetraspanins that arise from alternative splicing were recently identified at the mRNA level [50]. Remarkably, structural models predict many of these to have extracellular regions that are similar in architecture to conventional full-length tetraspanins, despite the existence of only one, two, or three transmembranes [50]. This would provide great flexibility for a single tetraspanin to be produced in differing forms to regulate a partner in a different manner, or to regulate an entirely different partner. However, this is highly speculative at present and studies are needed to establish whether non-conventional tetraspanins exist at the protein level and whether they have physiological relevance.

## 4. The TspanC8 Subgroup of Tetraspanins Regulate ADAM10

### 4.1. TspanC8 Overview

In 2012, we and others revealed that ADAM10 is specifically regulated by a subgroup of tetraspanins termed the TspanC8s, so called because they are unique amongst tetraspanins in containing eight cysteine residues within their large extracellular domain [51,52,53]. The subgroup consists of six members, Tspan5, Tspan10, Tspan14, Tspan15, Tspan17, and Tspan33, which share a relatively high degree of protein sequence identity. This ranges from 78% for the most related members, Tspan5 and Tspan17, to 26% for the most distantly related, Tspan10 and Tspan15. The TspanC8s interact with ADAM10 and promote its exit from the endoplasmic reticulum, enzymatic maturation, subcellular localization, lateral mobility in the plasma membrane and interactions with other membrane proteins (Figure 2) [51,52,53,54]. The interaction of TspanC8s with ADAM10 has been demonstrated in human cell lines, primary cells [52,55], mice [52,56], and *Drosophila* fruit flies [51]. An intimate association of TspanC8s with ADAM10 has been suggested by recent studies showing that not only does ADAM10 require TspanC8s for its exit from the endoplasmic reticulum, but Tspan5 and Tspan15 also have a heavy dependence on ADAM10 for their expression at the protein level [57,58,59]. Furthermore, co-immunoprecipitation experiments in cell lines with endogenous proteins have demonstrated that the majority of Tspan5 interacts with ADAM10 [59]. Thus, ADAM10 appears to exist as one of six TspanC8/ADAM10 scissor complexes.

A key question that arose following the discovery of TspanC8s as regulators of ADAM10 was whether different TspanC8/ADAM10 complexes have distinct substrate specificities. To address this, the shedding of known ADAM10 substrates was investigated in cell lines in which the TspanC8s were either knocked out using CRISPR/Cas9, knocked down using RNA interference, or over-expressed. The most definitive findings from these studies are that Tspan15/ADAM10 preferentially cleaves the cell–cell adhesion molecule neuronal (N)-cadherin [53,54,55], whereas Tspan5 and Tspan14 preferentially cleave and activate Notch [51,54,57,59]. This has led to the “six scissor” hypothesis, which proposes that ADAM10 is not a single scissor, but six different scissors with distinct substrate repertoires, depending on which of the six TspanC8s it is associated with [10,60,61,62].

### 4.2. Tspan15/ADAM10 Promotes Cell Invasion and Cleaves N-Cadherin

Tspan15 mRNA is expressed by many epithelial, endocrine, and endothelial cells, and by mucosal-associated invariant T (MAIT) cells and T helper 17 (Th17) cells (Figure 3) [63]. Tspan15 is also expressed at the protein level by human platelets [58], in which the combined quantitated protein copy number of the three expressed TspanC8s (Tspan14, Tspan15 and Tspan33) is similar to the copy number of ADAM10, suggesting a one-to-one TspanC8-ADAM10 interaction [10]. The interdependence of the Tspan15/ADAM10 interaction was recently demonstrated by an 80% reduction in Tspan15 protein expression levels in ADAM10-knockout cell lines [57,58]. Moreover, ADAM10 was the only substantial binding partner for Tspan15 identified by mass spectrometry following endogenous Tspan15 immunoprecipitation in the HEK-293 human embryonic kidney cell line under stringent detergent lysis conditions, and a synthetic ADAM10/Tspan15 fusion protein appeared fully functional [58]. Together, these data suggest that Tspan15 exists primarily as a Tspan15/ADAM10 scissor complex.

The function of the Tspan15/ADAM10 scissor has started to be investigated using mouse models and cancer cell lines deficient in or over-expressing Tspan15. The Tspan15-knockout mouse is viable and has no major phenotype, although N-cadherin shedding in the brain was reduced by approximately 80% [56], consistent with the earlier cell line data [53,54,55]. N-cadherin is well established as a pro-invasive cadherin, for example during the epithelial to mesenchymal transition that precedes cancer metastasis, during which epithelial (E)-cadherin is transcriptionally downregulated and N-cadherin upregulated. Tspan15 is upregulated in several cancers and was recently shown to promote cancer cell invasion in vitro and tumor cell formation in vivo [64,65,66], and N-cadherin shedding was shown to be impaired following Tspan15 knockdown in oral squamous carcinoma cell lines [64]. In a related study, ADAM10 was shown to promote in vitro invasion and in vivo tumor formation in mesothelioma [67], a rare but deadly cancer caused by asbestos exposure. Tspan15 was not investigated in this study, but is likely to be important because ADAM10 shedding of N-cadherin was proposed as the underlying mechanism, since recombinant soluble N-cadherin ectodomain rescued the invasion phenotype [67]. The authors propose that the soluble N-cadherin ectodomain functions by stabilizing fibroblast growth factor receptor (FGFR) cell surface expression and signaling, because the ectodomain’s rescuing effect was abrogated following inhibition of FGFR signaling [67], and N-cadherin’s functional interaction with FGFR is mediated by the N-cadherin ectodomain [68]. ADAM10 knockout in mouse embryonic fibroblasts results in increased plasma membrane localization of N-cadherin and associated β-catenin, and impaired β-catenin signaling [17], and evidence for reduced β-catenin signaling was also observed following Tspan15 knockdown in oral squamous carcinoma cell lines [64]. In summary, Tspan15/ADAM10 shedding of N-cadherin is likely to be an important regulator of its role in invasion via multiple mechanisms: release of the ectodomain to stabilize FGFR signaling, promotion of β-catenin signaling, and strength of cell adhesion by limiting N-cadherin surface expression levels.

A scissor-independent role for Tspan15 was recently reported to explain Tspan15′s role in promoting esophageal cancer cell invasion [66]. NF-κB signaling was strikingly promoted in Tspan15-over-expressing cells and impaired in Tspan15-knockdown cells. The authors propose that Tspan15 promotes NF-κB signaling by interacting with the E3 ubiquitin ligase beta-transducin repeat containing 1 (βTrCP1), which is involved in NF-κB signaling via recognition and degradation of the NF-κB inhibitor IκBα. Indeed, Tspan15′s pro-invasive effect was impaired following βTrCP1 knockdown [66]. However, βTrCP1 was not detected in mass spectrometry analyses of Tspan15 immunoprecipitates [58,65]; therefore, further studies are needed to confirm the Tspan15-βTrCP1 interaction and to determine whether it is direct or involves one or more bridging proteins. Taken together with the N-cadherin data, there are clearly multiple mechanisms to help explain the pro-invasive role of Tspan15. Additional possibilities may result from future identification of the full repertoire of Tspan15/ADAM10 substrates, and the relative importance of ADAM10-dependent and independent roles are likely to be cell-type-dependent.

### 4.3. Tspan5/ADAM10 and Tspan14/ADAM10 Cleave and Activate Notch

Tspan5 mRNA is expressed by many cell types, but with no obvious enrichment in any particular types of cells, whereas Tspan14 is similar to ADAM10 in being expressed in most cell types (Figure 3) [63]. The Rubinstein group showed that Tspan5 and Tspan14 are important for Notch signaling in cell line models [51,54,57,59]. These experiments were conducted in the U2OS bone osteosarcoma cell line stably transfected with Notch1 or wild-type HeLa epithelial cells. OP9 stromal cells stably transfected with Delta-like 1 (DLL1) were used to activate Notch. The overall conclusion from TspanC8 knockdown or over-expression is that Tspan5 and Tspan14 promote Notch signaling whereas Tspan15 and Tspan33 do not. Redundancy between Tspan5 and Tspan14 was demonstrated by the necessity to knock down both proteins for Notch inhibition in some experiments, and an inhibitory effect of Tspan5 antibodies was only observed in Tspan14-knockdown cells [59]. The Tspan5-knockout mouse is viable and has no reported phenotype [59], while the Tspan14-knockout mouse has yet to be reported. In future, it will be interesting to determine whether Tspan14 single and/or Tspan5/14 double knockout mice phenocopy the embryonic lethality of ADAM10 and Notch1/4 knockout mice, which would provide strong genetic evidence for the importance of these tetraspanins in Notch activation.

Notch signaling is important in promoting many cancers, so one prediction of the Tspan5/14/ADAM10 Notch activation model is that Tspan5 and Tspan14 will promote such cancers. The first publication to support this prediction recently demonstrated that Tspan5 expression is associated with poor prognosis in hepatocellular carcinoma [69], one of the most common and lethal cancers. In hepatocellular carcinoma cell lines, Tspan5 over-expression promoted cell invasion in vitro, and tumor formation in the lung in a mouse model, while Tspan5 knockdown had the opposite effect. Tspan5 expression correlated with Notch activation in these cell lines, and a γ-secretase inhibitor abrogated the invasion-promoting effect of Tspan5 over-expression [69]. To increase our understanding of potential Tspan5/14 redundancy in Notch activation, it will be interesting to know the relative expression levels of Tspan5 and Tspan14 in the hepatocellular carcinoma cell lines, and whether combined Tspan5/14 knockdown has an even stronger inhibitory effect than either alone.

To address potential redundancy amongst TspanC8s in Notch signaling, it will be useful to extend the above studies to include Tspan10 and Tspan17. Tspan10 is a relatively poorly characterized member of the TspanC8 group, and it is relatively weakly expressed at the mRNA level or absent from most cell types (Figure 3) [63]. It also appears to be restricted to intracellular vesicles in transfected cells [51]. However, there is evidence for a Tspan10 role in Notch signaling, since it is upregulated during the in vitro differentiation of mouse bone marrow macrophages to osteoclasts, and Tspan10 knockdown strikingly impairs osteoclast differentiation and Notch signaling [70]. In this study, Tspan5 was also shown to be upregulated in osteoclasts and its knockdown also impaired their differentiation [70], but other TspanC8s were not investigated. Finally, Tspan17 is worthy of study in relation to Notch signaling because Tspan17 and Tspan5 are the most highly related by protein sequence (78% identity), and Tspan17 and Tspan14 are also relatively highly related (57% identity), so all three may play a role in Notch signaling. Tspan17 mRNA is expressed by most cell types but at relatively low levels (Figure 3) [63]. The phenotypes of Tspan17 or Tspan10 knockout mice have yet to be reported.

It is unclear why multiple TspanC8s play a role in Notch signaling. This could reflect the fact that there are four Notch proteins and different TspanC8/ADAM10 complexes may have distinct Notch preferences. An additional possibility is that the different subcellular localization profiles of the different TspanC8/ADAM10 complexes may enable Notch signaling in different subcellular compartments, for example, intracellular cleavage of Notch by ADAM10 that has been proposed following the ubiquitination and endocytosis of ligand-activated Notch [71].

### 4.4. Additional Substrate Specificities for TspanC8/ADAM10 Complexes

The adhesion molecule vascular endothelial (VE)-cadherin appears to be a substrate for ADAM10 in complex with either of the two highly related TspanC8s, Tspan5 or Tspan17 [72]. These TspanC8s were shown to be important in primary endothelial cells for limiting VE-cadherin surface expression and facilitating T cell transmigration through endothelial monolayers in an in vitro model system of inflammation. The other TspanC8s were not involved in these processes [72]. In contrast, Tspan15 was required for VE-cadherin cleavage in HEK-293 cells transfected with VE-cadherin [58]. These data show how the TspanC8/ADAM10 substrate specificities can vary in different cell types, and emphasize the importance of studying substrate shedding in the most physiologically relevant systems.

Four studies that have addressed APP cleavage have not defined a specific TspanC8/ADAM10 scissor for this substrate [53,54,56,73]. Tspan15 over-expression increased APP cleavage in N2A mouse neuronal cells [53], but decreased APP cleavage in HEK-293 [73] and U2OS cells [54]. Tspan15 knockdown in PC3 human prostate cancer cells had no effect on APP cleavage [54], nor did Tspan15 knockout affect APP cleavage in mouse brains [56]. Furthermore, over-expression of Tspan5, Tspan14, or Tspan33 did not promote APP cleavage in U2OS cells, and Tspan5 knockdown had no effect [54]. Tspan15, however, may be important for cleavage of the neural glial-related cell adhesion molecule (NrCAM), because cleavage was increased following Tspan15 over-expression in HEK-293 cells [73]. Tspan15 may also be important for cleavage of cellular prion protein, because reduced cleavage was observed in Tspan15-knockout mouse brains [56]. Finally, Tspan5 may be involved in cleavage of the adhesion molecule CD44, because Tspan5 knockdown partially reduced CD44 cleavage in PC3 cells, but Tspan15 knockdown had no effect [54]. These described studies were all targeted to specific substrates. In future, to fully address the six scissor hypothesis and potential compensation between TspanC8s, it will be important to determine TspanC8/ADAM10 sheddomes using an unbiased approach, such as through mass spectrometry identification of shed substrates in wild-type versus TspanC8 CRISPR/Cas9 single and multiple knockout cell lines.

### 4.5. Potential Mechanisms for TspanC8/ADAM10 Substrate Specificity

It is currently unclear precisely how TspanC8s modulate ADAM10 substrate specificity. Potential mechanisms include regulating ADAM10 subcellular localization, direct interaction with substrates and modulation of ADAM10 conformation. The evidence for each of these possibilities will now be discussed in turn.

There is strong evidence that TspanC8s regulate ADAM10 subcellular localization. The most striking examples have used green fluorescent protein (GFP)-tagged TspanC8s transfected into cell lines. These studies have shown that the localization of Tspan10, and to a lesser extent Tspan17, is restricted to late endosomes of HeLa cells, relative to the other TspanC8s which are expressed more strongly at the plasma membrane [51]. In contrast, Tspan15 and Tspan33 are relatively strongly localized to the plasma membrane in mCCD mouse kidney collecting duct cells, relative to other TspanC8s that are more intracellular [74]. In the latter study, Tspan33 localized ADAM10 to apical junctions between cells, whereas Tspan15 directed ADAM10 to lateral junctions [74]. Furthermore, Tspan15 is more stably localized at the surface of multiple cell lines compared with Tspan5, and undergoes endocytosis with ADAM10 more slowly than Tspan5 [57]. Consistent with such data, comparisons of the interactomes for Tspan5 versus Tspan15, by co-immunoprecipitation in low stringency detergent lysates followed by mass spectrometry, identified ADAM10 as the major interactor but there were some differences in other tetraspanins and tetraspanin-interacting proteins identified [54]. In the same study, single particle tracking of endogenous ADAM10 showed a 50% increase in its diffusion coefficient following transfection of Tspan15, but not Tspan5, suggesting that Tspan15/ADAM10 complexes have greater lateral diffusion in the plasma membrane [54]. Despite these differences in TspanC8/ADAM10 subcellular localization and dynamics, there is currently no definitive evidence that substrate specificity is dictated by localization. However, the most interesting data in this regard derives from the recent characterization of an extensive range of Tspan5/Tspan15 chimeric proteins [57]. This study showed the Tspan15 cytoplasmic regions to be important for its stabilization at the cell surface, which correlates with Tspan15′s negative impact on Notch signaling compared with Tspan5. Therefore, it is possible that at the nanoscale level Tspan15/ADAM10 localization does not enable its access to Notch [57].

Direct TspanC8 interaction with substrates is a potential mechanism for dictating substrate specificity, but there is no clear evidence for this yet. Indeed, testing this idea is likely to be challenging because potential TspanC8-substrate interactions may be of low affinity and transient. Interestingly, CD44 shedding was found to be partially dependent on Tspan5 and was more strongly detected in immunoprecipitates of Tspan5 than Tspan15 by mass spectrometry, while the opposite was the case for the Tspan15/ADAM10 substrate N-cadherin [54]. It is possible that complementary hydrophobic and/or charged regions on the surface of TspanC8s and substrates may facilitate the appropriate transient interactions to enable the positioning of the ADAM10 scissor and substrate for optimal shedding. Live cell super-resolution microscopy techniques may enable this possibility to be tested in future.

There is some evidence that TspanC8s may modulate ADAM10 conformation which could in turn impact on substrate shedding. The crystal structure of the ADAM10 ectodomain revealed an arrowhead-type structure [75]. The cysteine-rich region is at the top, forming the tip of the arrow, followed by the disintegrin domain and the N-terminal metalloproteinase domain close to the plasma membrane [75], thus positioned to cleave its substrates at cut sites that are typically 5–15 amino acids up from the transmembrane/extracellular boundary. For example, the cleavage site is 5 amino acids from the end of the transmembrane region for the platelet collagen receptor GPVI [24], 9–10 amino acids for N-cadherin [76], epithelial (E)-cadherin [77], betacellulin and EGF [78], and 15 amino acids for Notch1 [6]. It is therefore possible that the angle of the metalloproteinase domain is slightly different when ADAM10 is in complex with different TspanC8s, to enable the optimal cleavage of differently positioned cut sites. Consistent with this theory, the different TspanC8s interact via their main extracellular regions with distinct regions of ADAM10, as shown using co-immunoprecipitation of the different TspanC8s with ADAM10 truncation constructs [55], thus suggesting that ADAM10 conformations could be modulated. However, the use of Tspan5/Tspan15 chimeric constructs to investigate Notch activation did not support this idea because replacing the main extracellular region of Tspan15 with that of Tspan5 did not confer this Tspan15 chimera with Tspan5′s Notch-activating capability [57]. Ultimately, high resolution cryo-EM structures of TspanC8s in complex with ADAM10 will be required to properly address this conformational hypothesis.

Finally, TspanC8 conformational changes, suggested by recent tetraspanin structural studies [38,39,40,41,46], could help to explain the poorly understood mechanism of ADAM10 activation. In the ADAM10 crystal structure, the active site of the metalloproteinase domain is partially blocked by the C-terminus of the cysteine-rich domain [75]. Alterations in the closed/open conformation of TspanC8s could transmit a conformational change to ADAM10′s cysteine-rich region, thus opening up ADAM10′s active site for shedding activity. This is highly speculative and we must first discover whether tetraspanins can truly function as molecular switches and how switching is regulated.

## 5. Concluding Remarks: Therapeutic Potential of TspanC8 Targeting

Therapeutic targeting of tetraspanins is in its infancy [79]. Current efforts in this area are centered on tetraspanin CD37 as a target for B-cell blood cancers. Treatment with DuoHexaBody-CD37 is in clinical trial phase I for B-cell non-Hodgkin lymphoma [80] (NCT04358458, www.clinicaltrials.gov (accessed on 21 May 2021)). DuoHexaBody-CD37 is a biparatopic antibody, which is a type of bispecific antibody that binds two distinct epitopes on the same target. It also has a point mutation within the Fc region to promote hexamerization, which results in enhanced tumor killing via complement-dependent cellular cytotoxicity. DuoHexaBody-CD37 also promotes killing via antibody-dependent cellular cytotoxicity and antibody-dependent cellular phagocytosis [81]. As an alternative CD37-targeting strategy, treatment with CAR-37, a chimeric antigen receptor T cell targeting CD37, is in clinical trial phase I for CD37-positive blood cancers [82] (NCT04136275, www.clinicaltrials.gov (accessed on 21 May 2021)).

Therapeutic targeting of TspanC8s offers the potential to target ADAM10 activity in a cell type- or substrate-specific manner, without the toxic side effects that are likely from targeting ADAM10 on every cell in the body. Targeting Tspan5 and/or Tspan14 has potential as an anti-Notch strategy for the treatment of Notch-driven cancers such as T-ALL and solid tumors. As discussed in the previous section, there is recent evidence that Tspan5 enhances Notch signaling to promote hepatocellular carcinoma [69], and a Tspan5 antibody has Notch inhibitory activity in a cell line model [59]. Anti-Notch therapies for cancer are the focus of numerous clinical trials, including a trial in phase II that will use a γ-secretase inhibitor to treat desmoid tumors, which are rare, slow growing, locally invasive tumors of soft tissues (NCT04871282, www.clinicaltrials.gov (accessed on 21 May 2021)). However, anti-Notch therapy suffers from dose-limiting toxicities, particularly in the gut [83]. It remains to be determined whether targeting Tspan5 and/or Tspan14 will provide effective Notch inhibition for certain cancers in which these proteins are over-expressed, and whether such a treatment will have lower toxicity than existing anti-Notch strategies such as γ-sectretase inhibition. Targeting Tspan15 also has potential as an anti-cancer strategy, given its pro-invasive role [64,65,66] and capacity to promote esophageal cancer growth in a mouse model [66], as discussed earlier. It will be important to understand Tspan15′s ADAM10-dependent role, for example via shedding of N-cadherin [53,54,55,56,64] or novel substrates, and ADAM10-independent role, via NF-κB signaling [66], in different cancers to realize optimal targeting strategies. The generation of Tspan15 mutant proteins that specifically impair ADAM10-dependent or ADAM10-independent functions may facilitate these studies.

Therapeutic antibodies are the fastest growing class of therapeutic agents and provide great potential for the modulation of tetraspanins, given that they are cell surface targets. Moreover, the recent structural breakthroughs in tetraspanin biology [38,40,41,46] have enabled us to imagine how tetraspanins might be inhibited by locking them into a particular conformation, by preventing interaction with a partner such as ADAM10, or by preventing a TspanC8/ADAM10 complex from engaging with a substrate. To achieve this, it may be useful to adopt antibody display technology to generate smaller antibody or nanobody reagents, which have the potential to access epitopes that larger, conventional antibodies cannot. Alternative approaches may also be possible. These include synthetic peptides from tetraspanins that have been shown to have functional activity, for example, peptides from the CD9 main extracellular region that inhibit bacterial adhesion, potentially via disrupting tetraspanin/partner interactions [84]. Small molecules also offer potential to target functional regions of tetraspanins, such as the intramembrane cavity shown to bind cholesterol [38], or to block tetraspanin interactions including membrane partners such as ADAM10 and intracellular signaling proteins such as βTrCP1 that is proposed to link Tspan15 to NF-κB signaling [66]. Finally, it would be therapeutically desirable to activate TspanC8/ADAM10 in some instances, for example, to induce cleavage of APP to treat Alzheimer’s disease. However, this presents a challenge regarding how to specifically induce activation, and may lead to off-target toxicity.

To conclude, recent evidence strongly suggests that ADAM10 exists as a scissor complex with one of six regulatory TspanC8 tetraspanins that dictate its substrate specificity. Future work should aim to understand the mechanisms that underpin this specificity and to fully identify which of the six TspanC8/ADAM10 scissors cleave which ADAM10 substrates. This may reveal new therapeutic targeting possibilities for the many human diseases in which ADAM10 is implicated, since targeting particular TspanC8s may limit toxicity and enable the inhibition or activation of ADAM10 in a relatively substrate- or cell-specific manner.

## Figures and Tables

**Figure 1 ijms-22-06707-f001:**
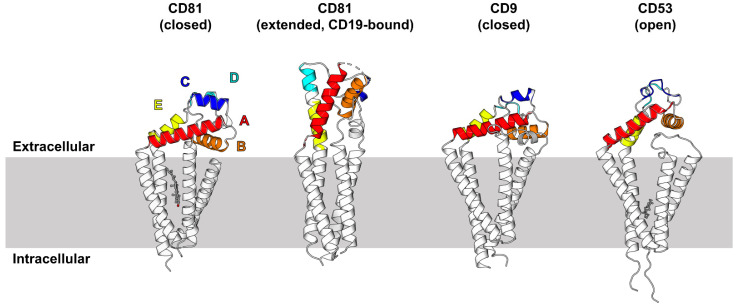
Tetraspanin structures. The helices/loops A–E in the main extracellular region are colored red, orange, dark blue, pale blue, and yellow, respectively. The lipid molecules identified in the intramembrane cavity are shown in grey, namely cholesterol for closed CD81 and monoolein for CD53. Protein Data Bank (PDB) IDs are 5TCX for closed CD81 [38], 7JIC for extended CD19-bound CD81 [39], 6K4J for CD9 [40], and 6WVG for CD53 [41].

**Figure 2 ijms-22-06707-f002:**
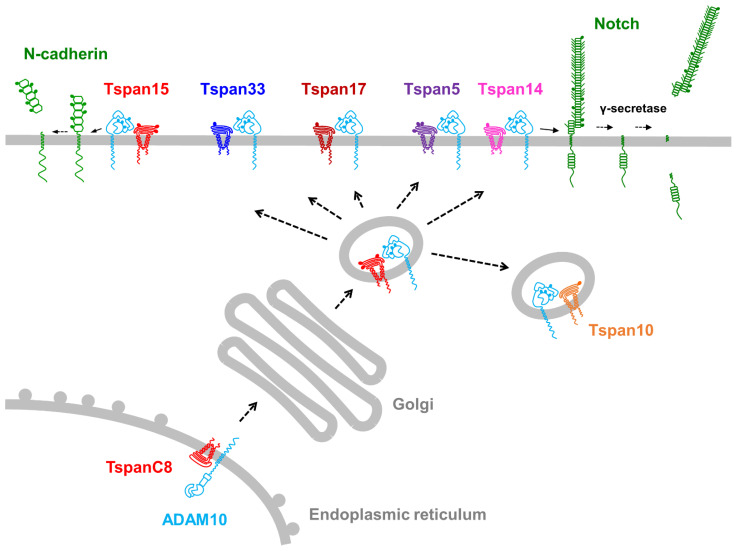
Regulation of ADAM10 by TspanC8 tetraspanins. ADAM10 interaction with one of six TspanC8 tetraspanins is required for its exit from the endoplasmic reticulum, trafficking to the cell surface or intracellular vesicles, and cleavage of distinct substrates. Definitive examples of the latter are cleavage of N-cadherin by Tspan15/ADAM10 complexes and cleavage of Notch by Tspan5/ADAM10 and Tspan14/ADAM10 complexes. Filled circles on membrane protein ectodomains represent N-linked glycosylation and angled lines represent O-linked glycosylation.

**Figure 3 ijms-22-06707-f003:**
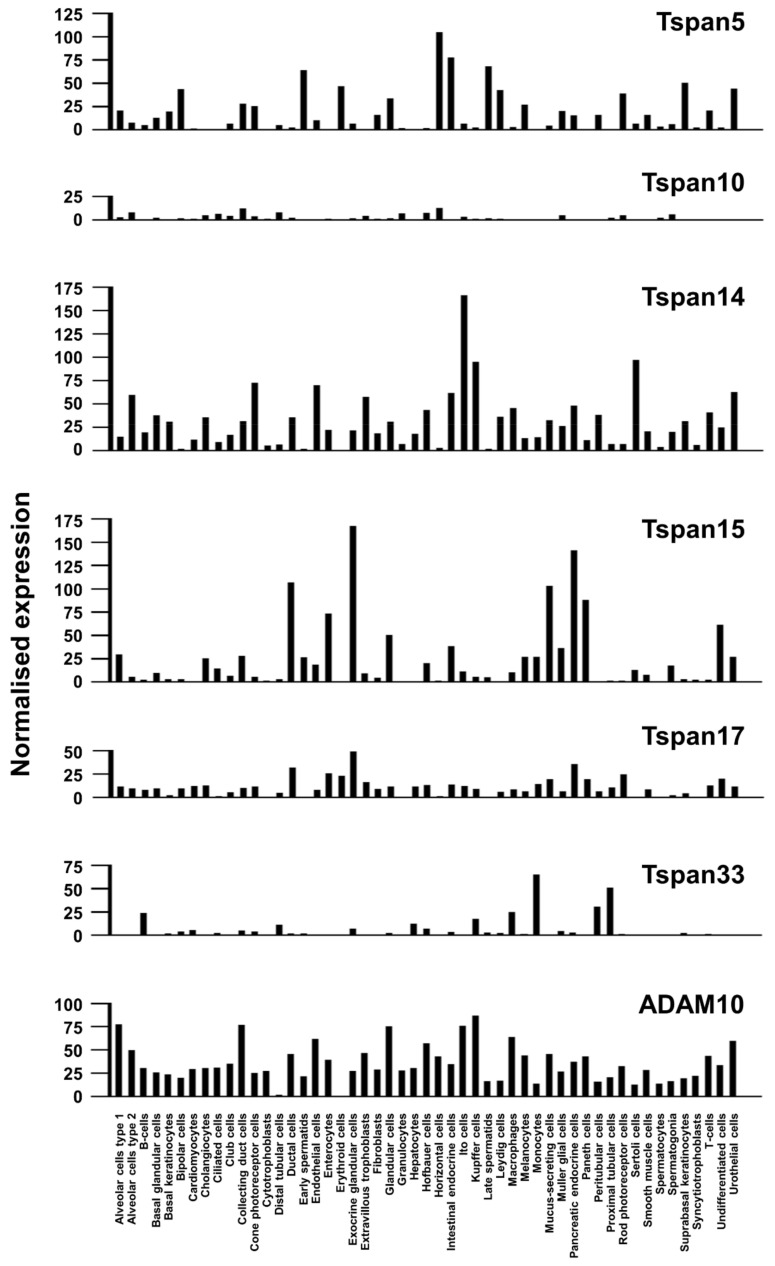
Expression profile of TspanC8s and ADAM10. The data is compiled from single-cell RNA sequencing data from the Human Protein Atlas (www.proteinatlas.org (accessed on 21 May 2021)) [63].

## Data Availability

Publicly available datasets were analyzed in this study. Structural data can be found in the Protein Data Bank (PDB) here: https://www.rcsb.org/ (accessed on 21 May 2021). Gene expression data can be found in the Human Protein Atlas here: https://www.proteinatlas.org/ (accessed on 21 May 2021).

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
