# Peer review of "Regulation of ADAM10 by the TspanC8 Family of Tetraspanins and Their Therapeutic Potential"

_ijms, 2021, doi:10.3390/ijms22136707_

Round 1

Reviewer 1 Report

  • The abstract of this review article has some self-plagiarism from their own previous publication (https://doi.org/10.1074/jbc.RA120.012601). Modification of copied part is recommended. 
  • The logic of this manuscript is easy to follow. They prepared this review article based on necessity from their research topic and previous publications. They clearly address this ADAM10 needs further investigation as a therapeutic target for many diseases, though the outcome of targeting this protein is completely different depending on tissues and disease models.  Addressing future study direction, they show interesting predictions with possible animal models for each substrate involved in the cleave action ADAM10. Therapeutic targets and potential approaches (antibodies, small molecules) from other studies are well summarized.
  • In line 142, they criticized previous reports in the literature using "co-immunoprecipitation which is prone to artefacts, lack the functional or structural validation", whereas in line 196 and line 242 they cited other reports, "consistent with previous co-immunoprecipitation data" and "co-immunoprecipitation experiments in cell lines". This is self-conflict. They cannot insist on this unless they have a better method than a well-accepted method in the protein interaction study area. Modification in line 142 is recommended. 

Author Response

We thank the reviewer for reviewing our manuscript and for their positive comments.

We are very grateful to the reviewer for noticing our self-plagiarism mistake.  We could not find any self-plagiarism with the article indicated by the reviewer (https://doi.org/10.1074/jbc.RA120.012601), so we ran the abstract text through SmallSEQTools, a free online plagiarism checker.  This detected that the first sentence of our abstract was exact self-plagiarism with our previous review article (https://doi.org/10.3389/fimmu.2018.01451).  To address this problem, we have modified the first sentence of the abstract and re-run the abstract through SmallSEQTools; there is no longer any indication of plagiarism.

We also appreciate the reviewer’s comment on our inconsistent criticism of co-immunoprecipitation.  To address this, we have modified the text around line 142 to the following: “There are many other tetraspanin/partner protein interactions reported in the literature, often detected by co-immunoprecipitation in detergent lysates. It will be important to assess the relevance of such interactions using the functional or structural validation approaches described for the examples above.”

Finally, while thinking about and acting on the comments of both reviewers, we realised that we had considerable text (about a page and a half) describing the five recent tetraspanin structural papers, but without any explanatory figure.  We now think that it is important to include a figure to enhance the reader’s understanding of these major developments in the tetraspanin field.  Indeed, our review manuscript is timely in being the first such review to discuss these recent developments.  We have therefore added a third figure to our manuscript, which becomes the new Figure 1.  This figure was generated by Chek Ziu (Connie) Koo in my group, and she has been included as an author.  We hope that the reviewer is happy with this addition.

Reviewer 2 Report

The review is focused on the regulation of MMP-10 (ADAM10) by subgroup of six  tetraspanins, the TspanC8 subgroup. 

Given that there are some controversies among the information presented, it would   be interesting to read a brief summary  after each section, with the main findings reported and with the opinion of the authors.  

In Conclusion, the authors could add some more perspectives  for future studies or suggested treatments. 

Author Response

We are grateful to the reviewer for their positive scoring of our manuscript.

The suggestions for improvement from the reviewer were as follows: “Given that there are some controversies among the information presented, it would be interesting to read a brief summary after each section, with the main findings reported and with the opinion of the authors. In Conclusion, the authors could add some more perspectives for future studies or suggested treatments.”

To address these comments, we hope that the new structural figure (new Figure 1 – see paragraph below) has provided extra clarity on the differences in structure between the three tetraspanins that have been reported.  We also hope that our modification around line 142, as suggested by the other reviewer, has helped to clarify the issue around co-immunoprecipitation.  In the Conclusion, we have added the following extra sentence at about line 500: “The generation of Tspan15 mutant proteins that specifically impair ADAM10-dependent or ADAM10-independent functions may facilitate these studies.”  But we do feel that there is already a substantial amount of opinion and future experiment suggestions in this section.  Also, we feel that adding a brief summary at the end of each section would create repetition that would not be particularly helpful, particularly given that we have informative subheadings throughout that summarise the take-home messages of each section.

Finally, while thinking about and acting on the comments of both reviewers, we realised that we had considerable text (about a page and a half) describing the five recent tetraspanin structural papers, but without any explanatory figure.  We now think that it is important to include a figure to enhance the reader’s understanding of these major developments in the tetraspanin field.  Indeed, our review manuscript is timely in being the first such review to discuss these recent developments.  We have therefore added a third figure to our manuscript, which becomes the new Figure 1.  This figure was generated by Chek Ziu (Connie) Koo in my group, and she has been included as an author.  We hope that the reviewer is happy with this addition.